# Factors Affecting English Language Teachers’ Behavioral Intentions to Teach Online under the Pandemic Normalization of COVID-19 in China

**DOI:** 10.3390/bs13080624

**Published:** 2023-07-27

**Authors:** Yanjun Gao, Su Luan Wong, Mas Nida Md. Khambari, Nooreen bt Noordin, Jingxin Geng, Yun Bai

**Affiliations:** 1School of Foreign Languages, Anyang Institute of Technology, Anyang 455000, China; gs56279@student.upm.edu.my; 2Department of Science and Technical Education, Faculty of Educational Studies, Universiti Putra Malaysia, Serdang 43400, Malaysia; 3Department of Foundations of Education, Faculty of Educational Studies, Universiti Putra Malaysia, Serdang 43400, Malaysia; khamasnida@upm.edu.my; 4Department of Language and Humanities Education, Faculty of Educational Studies, Universiti Putra Malaysia, Serdang 43400, Malaysia; nooreen@upm.edu.my (N.b.N.); gs56324@student.upm.edu.my (J.G.); 5School of Foreign Languages, Cangzhou Normal University, Cangzhou 061000, China; yun_bai@caztc.edu.cn

**Keywords:** intentions, English language teachers, online teaching, pandemic normalization, technology acceptance model

## Abstract

The unexpected outbreak of COVID-19 pandemic forced most teachers across the globe to switch their teaching from traditional face-to-face to online without having received adequate preparation and knowledge related to online teaching. To better comprehend teachers’ willingness to conduct emergency remote teaching during the worldwide crisis, the current study was designed to examine teachers’ intentions and, in particular, the factors affecting their behavioral intentions to teach online, by contextualizing the research in the English language teaching settings in China. The research model was developed based on an extended technology acceptance model (TAM) by adding subjective norm, self-efficacy, technological complexity, and facilitating conditions into the original TAM. The data were obtained from a total of 287 respondents including 228 (79.4%) female teachers and 59 (20.6%) male teachers via online questionnaires. The questionnaire was developed by adapting previously validated instruments and then refined by two educational technology experts in order to better suit the current study. The empirical findings, using structural equation modeling (SEM), showed that the extended TAM was valid in predicting English language teachers’ intentions to teach online during the pandemic normalization. At the same time, the findings suggested that teachers’ intentions were influenced significantly by attitude, facilitating conditions, and self-efficacy; language teachers’ attitude toward online teaching was significantly affected by both its perceived usefulness and perceived ease of use; perceived usefulness had a significant association with perceived ease of use and subjective norms; and perceived ease of use was significantly impacted by self-efficacy and facilitating conditions. Perceived usefulness was not suggested as a significant determinant of English language teachers’ intentions to adopt online teaching. Despite some limitations, the study has several implications from the perspective of theory and practice. The more factors with a higher influence should be determined from different perspectives in future research.

## 1. Introduction

The high infectivity of Corona Virus Disease 2019 (COVID-19) in human-to-human transmission forced China’s epidemic prevention and control efforts into normalization. Normalization indicates that epidemic prevention and control work in China will be taken as a long-term strategic task until production and life order is restored to normal [1]. In a certain sense, online teaching may become a long-lasting and preferred delivery mode during this crisis [2]. Such a special period rendered online teaching a must for ensuring educational continuity, rather than a volitional option for improving educational quality [3]. For Chinese English teachers, it was also the first time relying on various online platforms, devices and the Internet as the only delivery media to present their teaching assignments [4]. Due to their inadequate preparation for the emerging phenomenon of remote teaching, English language teachers faced new challenges in developing their online instructional skills in parallel with standard online courses [5]. Consequently, some English language teachers showed negative attitudes toward, or even refused to conduct, online teaching, which would definitely deteriorate online education quality [6].

Whenever technology is used in conjunction with education, the teacher is one of the central players [7]. In other words, in educational settings, teachers’ willingness can significantly influence the integration of technology into teaching practice. Previous research, however, has indicated that teachers were reluctant to accept different forms of online teaching due to a variety of reasons, such as concerns regarding the reliability of online teaching, the fear of unpredictable changes, and worries about workload issues [8,9]. In China, although previous studies have focused on English online teaching during the COVID-19 pandemic from different perspectives, such as English teachers’ anxiety during livestreaming [10] and teachers’ role in online settings [11], few studies have discussed EFL teachers’ intentions regarding online teaching during this confinement period. Therefore, an evidence-based evaluation of English language teachers’ behavioral intentions in adopting online teaching during the pandemic normalization process is important. Furthermore, it is unclear which factors have an impact on English language teachers’ adoption of online teaching, and to what extent these factors influence the adoption process under normalized regulations. Therefore, it is essential to devote scholarly attention to identify the influential factors for EFL teachers’ intention to adopt online teaching.

In recent decades, studies about individual’s intentions to use technology have been carried out via modeling psychological constructs [12]. A widely employed and validated among these theories and models is the technology acceptance model [13]. As a predictive model for understanding the user’s behavioral intentions regarding the use of technologies, TAM [13] has gained acknowledged popularity and is regarded as a key model [14], the best model [15], a powerful and robust one [16], or the gold standard [17]. The TAM was developed for the purpose of identifying fundamental variables recommended by previous studies that explained the technology acceptance of users from the perspective of cognition and affection. The model specifies the relationships among perceived usefulness (PU), perceived ease of use (PEU), attitude (ATU) towards technology use and behavioral intention (BI) regarding technology use. Overall, the TAM has received empirical support as a valid model to understand the technology acceptance of users in various contexts [18].

However, the TAM has been criticized for decades, mainly due to its over-simplicity, which lowers its explanatory power and provides less insight into technology acceptance [19]. Therefore, there have been calls to extend the model by including other constructs [20], so that the technology acceptance can be better revealed from wider perspectives. In response to these suggestions and criticisms of its too being parsimonious, the extended TAM models, adding some external variables, were proposed and validated. TAM research indeed continues to evolve and develop, but some external constructs such as self-efficacy (SE), facilitating conditions (FC), subjective norms (SN), and technological complexity (TC) and the relationships with those specified in the TAM, have remained relatively stable [7,12].

For the current study, by extending TAM [13] with the salient constructs, the purpose of the study is to investigate what factors will have an influence on English teachers’ behavioral intentions to adopt online teaching under the normalization of the COVID-19 pandemic. Specifically, the following questions will be examined:

RQ1: What are the significant relationships among the selected factors (i.e., English teachers’ attitude, perceived usefulness, perceived use of ease, subjective norm, self-efficacy, technological complexity, and facilitating conditions) in the extended TAM that affect English teachers’ intentions to teach online under the pandemic normalization?

RQ2: What is the contribution of the selected factors in the extended TAM to explain English teachers’ intentions to teach online under the pandemic normalization?

The study hopes to contribute to the existing literature on technology acceptance by applying extended TAM to Chinese EFL teachers’ intentions to adopt online teaching under the normalization of the COVID-19 pandemics. It is hoped that the findings can also generate insights into the factors that affect EFL teachers’ intentions regarding online teaching. Finally, the study’s conclusions may serve as a guide for those involved in the field of education on how to take more efficient measures to improve EFL teachers’ intentions to implement online teaching, especially in times of crisis.

## 2. Literature Review

TAM was first defined by David [13] as a theory that explains the factors influencing the intention to use information technology to improve performance in organizations. In addition to the behavioral intention to use information technology, TAM was further applied to a technology-friendly learning environment and learning management systems in education settings [21,22]. Therefore, in the current study, TAM was applied to the Chinese EFL teaching context from the perspective of EFL teachers. Meanwhile, some factors (e.g., individuals, application characteristics, belief, application characteristics, technology, and social influence) were found to be significant determinants of users’ behavioral intentions to use technology in education settings [2,4,23,24,25]. Therefore, in the study, external variables based on previous findings were added to extend the TAM to offer it a better explanatory power for addressing the acknowledged shortcomings of TAM regarding its being parsimonious. These external constructs included individual factors such as self-efficacy, technological factors such as technological complexity, and organizational and social factors such as subjective norm and facilitating conditions. A detailed description of these added external variables, as well as the variables specified in the TAM, are presented as follows.

### 2.1. Technology Acceptance Model (TAM)

Technology acceptance research focuses primarily on users’ willingness to use technology per se. Based on findings from technology acceptance research originally conducted within business and information system contexts [26], several models have been proposed to determine key determinants that impact users’ intentions to accept technology. Among popular models in technology acceptance research, TAM [13] has been highlighted as a robust and powerful predictive model [27]. Theoretically, the Theory of Reasoned Action (TRA) proposed by Ajzen and Fishbein [28] serves as the theoretical foundation for TAM, which shows that people’s attitudes influence their social behavior. TAM was developed from TRA by extending and formulating it to identify fundamental variables that occur in technology acceptance at both cognitive and affective levels. Specifically, TAM outlines the relationships among variables including ATU, PEU, PU, BI, and actual usage (AU). PEU is hypothesized to directly affect PU. Together, PU and PEU have a direct impact on ATU which, in turn, influences BI directly, along with PU. Finally, AU is directly determined by BI. TAM is presented in Figure 1.

Since its development, TAM has been widely used to explain technology acceptance in diverse domains, such as in the automotive industry [29] and telemedicine [30]. In an educational context, TAM also has been widely verified in studies on users’ acceptance of technology. For instance, TAM was used to investigate teachers’ usage behavior of learning management systems [23,31,32] and measure students’ use of the Zoom application in a language course [33]. In terms of online education, previous research has demonstrated that TAM is an effective mode to explore technology acceptance in online settings [34,35]. Additionally, TAM is reliable in assessing language teachers’ intentions to employ technology in the context of language instruction [12,36].

Despite the widespread acceptance of TAM, there have been calls to extend the model to address more sophisticated relationships. It was recommended to study TAM further to gain larger insights into its validity [37]. In response to these suggestions and criticisms of its being too parsimonious, various extended TAM models, by adding some external variables, were proposed and validated. TAM research indeed continues to evolve and develop, but some factors (i.e., BI, ATU, PU, PEU, self-efficacy (SE), facilitating conditions (FC), subjective norms (SN), and technological complexity (TC)) and the relationships between them have remained relatively stable [7,12].

### 2.2. Research Model and Hypotheses

The current study is designed to develop and validate a comprehensive model to examine the factors that influence English teachers’ intentions to teach online under the pandemic normalization of COVID-19. Therefore, an extended research framework based on TAM was developed to guide the study, which synthesized the core constructs in the TAM and some salient external variables that have been found to influence the core variables of TAM. Finally, the extended TAM includes eight constructs to examine English language teachers’ intentions to adopt online teaching, including four core constructs—BI, ATU, PEU and PU—and four external constructs—SN, SE, TC, and FC.

#### 2.2.1. TAM Hypotheses

Based on the original TAM, the acceptance model consists of four constructs: BI, ATU, PEU and PU. ATU, one of the core variables in the TAM, refers to individual’s positive or negative feelings toward the target behaviors [13]. In the current study, ATU means the degree to which English teachers have negative or positive feelings toward online teaching. In addition, PU and PEU are highly significant constructs, since PU is a measure of a user’s belief that the use of technology will improve their productivity [13] and PEU refers to a user’s belief that technology use will be free of effort [13]. In this study, PU refers to the English language teachers’ belief that adopting online teaching will be useful and PEU refers to English teachers’ perception of not much effort being required in their online teaching practices. BI is initially proposed as a direct determinant of actual usage behavior, which indicates an individual’s readiness to conduct a specific task [15]. In this study, BI refers to the extent that English language teachers are willing to conduct online instruction. Altogether, TAM specifies the relationships between BI, ATU, PEU and PU, i.e., ATU and PU are identified as immediate antecedents of BI and PU and PEU, and jointly and directly associated with ATU, and PEU is hypothesized to directly affect perceived usefulness.

The evidence from previous research has shown that ATU has a strong correlation with BI in some empirical research [38,39]. In particular, it has been found that teachers’ attitude toward technology use tends to significantly affect their intentions to use technology in their teaching practice [40]. Previous research has also demonstrated that PU affected the BI of utilizing technologies [22,41,42]. In addition, the indirect relationship between PEU and BI through PU has been supported in the online learning environment [43,44]. Studies also have shown that teachers’ attitudes toward technology adoption are significantly influenced by PEU and PU [43,44]. Based on the model and previous research, TAM was used in the study by focusing on the relationships among the variables to examine the behavioral intentions that influence English language teachers’ adoption of online teaching. Therefore, the following hypotheses were established:

**H1:** 
*English language teachers’ attitudes toward online teaching will significantly influence their behavioral intentions to adopt online teaching.*


**H2:** 
*English language teachers’ perceived usefulness will significantly influence on their behavioral intentions to adopt online teaching.*


**H3:** 
*EF teachers’ perceived usefulness will significantly influence their attitudes toward online teaching.*


**H4:** 
*English language teachers’ perceived use of ease will significantly influence their attitudes toward online teaching.*


**H5:** 
*English language teachers’ perceived use of ease will significantly influence their perceived usefulness.*


#### 2.2.2. Subjective Norm (SN)

SN assesses a person’s perception of the opinions of others who are important to them regarding whether or not they should engage in particular behaviors [45]. As a manifestation of social influence to explain a person’s intentions to carry out a given action, SN was incorporated into the TRA [45] and, later, the theory of planned behavior (TPB) [46,47]. Despite the fact that the original TAM did not include SN due to theoretical and measurement issues, SN was later incorporated into TAM2 because Davis recognized the need for further research on the circumstances and mechanisms by which social factors affect usage behavior [13].

SN was hypothesized as a significantly direct determinant of BI and PU [46,48], which was supported by large effect sizes in a meta-analysis that investigated subjective norm in TAM conducted by Schepers and Wetzels [49]. The immediate influence of BI on SN revealed that an individual’s involvement in certain behavior can be influenced by the opinions of others who are thought to be important referents to them, even if they are not positive regarding the behavior or its consequences [26,50].

In the field of education, the reference group for teachers may include administrators, colleagues, students, and institutional goals and policies. Studies indicated that administrators played critical roles in determining the faculty’s work and the adoption of online teaching technology, and they created a strong SN to encourage or discourage the faculty to engage in online activities [8]. Other studies showed that institutional goals and policies affected the faculty’s adoption of technology [51]. Furthermore, colleagues affect an individual’s decision to use the system. When colleagues think a system is useful, the individual is more likely to think so as well [47]. In this study, it is known that China is a collectivist-cultural country. That is being said, it is very likely that the BI and PU of English language teachers would be influenced by their leaders, peers, students, and surrounding policies. As a result, considering the importance of SN in developing BI toward technological adoption and acceptance, as well as the effects it has on PU, the following hypotheses are established:

**H6:** 
*Subjective norms will significantly influence English language teachers’ behavioral intentions to adopt online teaching.*


**H7:** 
*Subjective norms will significantly influence English language teachers’ perceived usefulness of online teaching.*


#### 2.2.3. Self-Efficacy (SE)

Self-efficacy is an individual’s belief in her/his ability to cope with diverse conditions and successfully arrange and accomplish tasks [52]. Self-efficacy is a self-assessment and aids in a better understanding of human performance of a given task [53]. Teachers’ self-efficacy is defined, in the context of education, as their perceived confidence in their capacity to successfully carry out instructional activities [54]. Teachers that have high self-efficacy tend to have excellent relationships with their students, be skilled at handling difficulties in the classroom, show more commitment to their job, and actively incorporate technology into their teaching methods [3].

Reviewing the literature, the effect of SE on other variables is different. The study conducted by Teo et al. found that SE was a significant determinant that influenced pre-service teachers’ BI to accept technology and PEU [18]. There existed a positive influence of computer self-efficacy upon PEU and the acceptance decision of individual English teachers [23]. In the study of Mei et al., self-efficacy was suggested to have a direct relationship with PEU, but was not suggested to affect BI in the context of preservice teachers’ computer-assisted language learning [12].

Teachers’ self-efficacy in the context of the current study refers to the belief that English language teachers have in their ability to carry out online instruction. In general, it is anticipated that English language teachers with greater levels of self-efficacy will be more open to accepting online instruction and will view it as requiring less effort than those with lower levels of self-efficacy. As a result, in line with some earlier studies, the following hypotheses are established:

**H8:** 
*English language teachers’ self-efficacy will significantly influence their behavioral intention to adopt online teaching.*


**H9:** 
*English language teachers’ self-efficacy will significantly influence the perceived use of ease of online teaching.*


#### 2.2.4. Technological Complexity (TC)

The degree to which technology is seen as being relatively difficult to understand and operate is referred to as technological complexity (TC) [55]. The construct is intended to investigate the impact of technical factors on users’ perceptions of task ease. According to previous research, the influence of TC on the PEU was shown to be inconsistent in different studies. The study conducted by Teo et al. examined pre-service teachers’ technology acceptance and found a significant and negative relationship between TC and PEU [18]. However, there existed a strong and positive relationship between the two constructs regarding teachers’ adoption of learning management systems [21]. Nevertheless, Huang et al. did not find a significant effect of the TC on the PEU when they examined English teachers’ intentions to teach online in the mandated environment [5].

Chinese English teachers’ non-positive attitude toward combining technology with their teaching was attributed to the complications surrounding technology [56]. Given that TC may be a potential barrier to English teachers’ adoption of online teaching in the current research, the following hypothesis was established:

**H10:** 
*Technological complexity will significantly influence English teachers’ perceived ease of use of online teaching.*


#### 2.2.5. Facilitating Conditions (FC)

The degree to which a person feels that an organization and technological infrastructure exist to facilitate the use of the system is defined as the facilitating condition [48]. This is individual perception of external support, including resource-facilitating conditions such as policy support, money and time, and technology-facilitating conditions such as technical instructions and an accessible network [50]. FC is considered an important construct in UTAUT to predict an individual’s actual use of technology [48].

External facilitating conditions play an important role in teachers’ integration of technology into their teaching. In addition to the important effect of FC on BI, FC exerted an indirect influence on BI through PEU in the studies of teachers’ adoption of technology [23,57]. Furthermore, the study on Chinese language teachers’ perceptions of technology use in Hong Kong discovered that FC had a significant effect on PEU but not on PU [58].

In the current study, FC will be measured by the English teachers’ perception of whether they can access the required resources and necessary support to adopt online teaching. It is critical to determine whether the presence and lack of facilitating conditions have an impact on English language teachers’ adoption of online teaching; after all, technical support and technological conditions are essential to ensuring the successful implementation of online teaching, especially in the context of crises. Therefore, the following hypotheses are established:

**H11:** 
*Facilitating conditions will significantly influence English teachers’ behavioral intentions to adopt online teaching.*


**H12:** 
*Facilitating conditions will significantly influence English teachers’ perceived ease of use of online teaching.*


Building on this statement, the following research model (Figure 2) was proposed to examine English language teachers’ intentions to adopt online teaching.

## 3. Methodology

### 3.1. Research Design

The study is a quantitative study with the intent to investigate the relationship among variables such as PU, PEU, ATU, SN, SE, TC, FC and BI. Moreover, structural equation modeling (SEM) is used to examine the path relationships among these variables.

### 3.2. Sample Size and Sampling Technique

The target population for the study consisted of a total of 2235 English teachers in 29 public colleges in Henan province in China, who enrolled during the second semester of the 2021–2022 academic year. Based on Cochran’s calculation [59], a minimum sample size of 239 was determined for the study.

In order to guarantee that the target population was represented as accurately as possible, proportional stratified cluster sampling was used. The first step was to decide the stratum. In this study, university type was taken as a stratum; therefore, 29 public universities were divided into normal university, comprehensive university and the university of science and technology. Then, the proportion of the sample size and the number of respondents that should be selected from each stratum or subgroup (the university type) was calculated according to the size of each stratum. After the stratified proportion was established, the sample participants were chosen using a random cluster-sampling procedure with a fishbowl approach, which picked samples at random from established grounds or clusters. Each university within the stratified type of universities served as the cluster unit for this study. Finally, six universities were determined as the cluster to administrate the questionnaires. A total of 317 online questionnaires were returned, thereby meeting the minimum requirement of a sample size of 239, as suggested by Cochran [59]. Thirty questionnaires were deleted because either the respondents’ answering time was less than 200 s, incomplete information was provided, or the same answers were chosen for all the questions, resulting in the reduction in the final sample count to 287.

Among the respondents, 228 (79.4%) were female teachers and 59 (20.6%) were male teachers, indicating that female teachers comprised the majority of English language teaching in this study. In general, a greater proportion of respondents were aged from 36 to 45 years (47.4%). In terms of teaching experience, less than one-third of the teachers had less than 10 years of experience (28.9%) and two-thirds had more than 10 years. Nearly half of the respondents were lecturers (45.3%) and most teachers were Master’s degree holders (78.4%). The profile of 287 respondents is presented in Table 1.

### 3.3. Instrument

For this study, a two-section online questionnaire was developed. The Section 1 was self-reported demographic information, which included teacher’s gender, age, teaching experience, academic title, and highest degree. Eight construct scales from previously validated instruments made up the Section 2 (shown in Table 2). Each item on the survey was scored on a Likert scale from 1 to 5, representing answers ranging from strongly disagree to strongly agree. With high Cronbach’s alpha coefficients ranging from 0.833 to 0.972, all the original constructs were demonstrated to be internally consistent. The adapted items of the scales are listed in Appendix A.

Two educational technology experts refined all the instruments and ensured that no important components were overlooked based on the current online teaching context. Pre-testing with 5 English language teachers [61] was carried out to avoid ambiguity in the questions so that respondents could understand the questions the way they were intended. A pilot study with 30 respondents [62] was also conducted to ensure the actual study was carried out successfully. The feedback and suggestions from the pre-testing and pilot study were taken into consideration when further refining the questionnaire. Based on the results of the pilot study, the reliability of the scales was acceptable (shown in Table 2). In terms of language, given that all the respondents were English teachers who had high levels of English proficiency, the English version of the questionnaire was used in both the pilot and the actual study.

### 3.4. Data Collection

Prior to data collection, the application for ethical clearance for the research was approved by the Ethics Committee for Research Involving Human Subjects of the University. Due to the influence of the COVID-19 pandemic, as well as the advantages of using an e-questionnaire, an online questionnaire by Survey Star, a professional online questionnaire collection platform in China, was administered to the English teachers for data collection. The goal of the questionnaire and the importance of the respondents’ truthful participation and submission of the questionnaire were explained to the respondents. Within a two-week period, teachers could complete the questionnaires as they chose, including by phone, laptop or computer, at any time. To maintain the objectivity of the data, one submission requirement was established: the same questionnaire could only be submitted once from the same IP address. It was important to stress to the participants that their participation was voluntary and that they could leave the study at any time. Another extremely crucial point that was emphasized to all teachers was that their responses would be anonymized and all the data would be used solely for the research.

The data collection process took place over three weeks in July 2022. A total of 317 online surveys were completed and returned. After removing 30 invalid surveys, 287 questionnaires were left for the preliminary analysis, which satisfied Cochran’s recommendation [59] for the study’s minimal sample size of 239 questionnaires.

### 3.5. Data Analysis

All the data were analyzed by SPSS 25 and AMOS 24 for the study. First, a preliminary data analysis was carried out to check for missing data, normal distribution, outliers, and multicollinearity, followed by descriptive statistics.

Then, the claimed correlations were tested using structural equation modeling (SEM) (Figure 2). SEM was used because it can estimate measurement errors while concurrently analyzing integrated correlations between latent and observed variables and the relationships between latent variables. This leads to a more accurate measurement of the survey’s items and structures. According to Anderson and Gerbing’s advice [63], a two-step SEM procedure—the measurement model and structural model—was used in this study. The measurement model validated the relationships between the observable indicators and the underlying constructs, while the structural model examined the hypothetical relationship and determined the relationships between the latent variables presented in the model.

## 4. Findings

### 4.1. Preliminary Analysis

First of all, there were no missing data for the study. In addition, the assumption of the univariate normal was met. Specifically, the skewness indices of all items (ranging from −0.741 to 0.209) were within the criteria between −2 and +2 [64], and kurtosis indices of all the items (ranging from −0.516 to 0.929) were between −7 and +7 [65]. The multivariate normality of the observed variables was examined by Mardia’s normalized multivariate kurtosis value [66]. According to the data, Mardia’s coefficient was 131.486, which was less than the suggested value (P [p + 2]) of Raykov and Marcoulides [67], where P represents the total number of observed variables (24 [24 + 2]) = 624. In this regard, the assumption of multivariate normality was also met in the study. As for the detection of outliers, the Mahalanobis Distance Test was used. The distance value of Mahalanobis d-squared between samples was small (ranging from 0.095 to 4.204), indicating that no multivariate outliers were identified in the current dataset from the perspective of a practical suggestion. Finally, all the tolerance values of factors (ranging from 0.343 to 0.748) were bigger than the cut-off threshold of 0.10 and all their variance inflation factor (VIF) values (ranging from 1.336 to 2.918) were smaller than the cut-off threshold of 5. According to Hair et al. [68], these data did not have multicollinearity problems.

### 4.2. Descriptive Statistics

The mean, standard deviation, skewness, and kurtosis of eight constructs in the proposed research model were examined. First, all mean scores (ranging from 3.00 to 4.03) were higher than the mid-point of 3.00, which illustrated an overall positive response to the selected factors. The standard deviations (from 0.58 to 0.79) showed a narrow spread in respondents’ replies. Skewness indices (from −0.539 to 0.068) and kurtosis indices (from −0.476 to 0.715) were acceptable based on the recommendations of Kline [69], which were considered to be univariate normal.

### 4.3. Evaluation of the Measurement Model

For this part, model fit of measure model was examined first, and then confirmatory factor analysis (CFA) was conducted to validate the correction between items and factors. To be more specific, the convergent validity and discriminant validity, two important components of CFA, were used to assess the measurement model using the maximum likelihood estimation (MLE) procedure, which is regarded as a robust method in SEM [70].

#### 4.3.1. Model Fit of Measurement Model

Model fit was suggested to use several fit indices to measure [71], such as the Chi-square value (CMIN), degree of freedom (df), the ratio of CMIN and its degree of freedom (CMIN/df), standardized root mean square residual (SRMR), root mean square error of approximation (RMSEA), comparative fit index (CFI), and Tucker–Lewis’s index (TLI). Based on the generally recommended criteria (shown in Table 3), the fit indices of this study are indicative of a good fit of the measurement model.

#### 4.3.2. Convergent Validity

According to Fornell and Larcker [72], item reliability, composite reliability (CR), and average variance extracted (AVE) were utilized to evaluate the convergent validity of the measure items in this study. Item reliability shows how well an item explains its underlying construct [73]. A factor loading of 0.7 and above for each item is suggested [73] and, accordingly, the square multiple correlations (SMC) are ideal, with 0.5 or above. In this study, the values of the factor loadings of all items ranged from 0.702 to 0.866, and the lowest value of SMC was 0.504, indicating that each item had good reliability (shown in Table 4). For the CR, this means the internal consistency of all items of a construct. An acceptable value of 0.7 was recommended [72]. All the CR values of the factors in the study were higher than 0.8, which indicated the good internal consistency of a factor. AVE, as an important indicator of convergent validity, was used to measure the overall amount of variance in the items generated by the construct. The acceptable value of AVE is 0.5 or higher [72]. All the values of AVE of all the constructs ranged from 0.602 to 0.719 (shown in Table 4), showing the adequate convergent validity of each construct.

#### 4.3.3. Discriminant Validity

Discriminant validity indicates that a measure should not be correlated with another measure [74]. According to Fornell and Larcker [72], a given construct is more strongly associated with its indicators than the other constructs in the model if its square root of AVE is greater than the correlations of all other constructs. In this investigation, the correlations of variables off the diagonal were less than the square root of AVE for each construct on the diagonal (shown in Table 5), which indicated that the selected constructs were adequate in terms of their discriminant validity.

### 4.4. Evaluation of Structural Model

#### 4.4.1. Model Fit of Structural Model

As a whole, the research model had a good fit (shown in Table 6). All fit indices met the recommended acceptance level of structural model fitness, except for SRMR, which was a little higher (0.092) than 0.08.

#### 4.4.2. Tests of Hypotheses

Nine hypotheses out of twelve were supported (i.e., H1, H3, H4, H5, H7, H8, H9, H11 and H12), while three were not supported (i.e., H2, H6 and H10). Among the original variables in the TAM, BI was significantly influenced by ATU (H1) but not by PU (H2). PU and PEU were two important drivers of ATU (H3 and H4). PU was significantly affected by PEU (H5). In terms of the effect of external variables out of TAM, SN had no significant influence on BI (H6) but had a significant influence on PU (H7). SE had a significant influence on BI (H8) and PEU (H9). The effect of TC on PEU was negative but not significant (H10). Both BI and PEU significantly influenced by FC (H11 and H12). All the path results of the research model are presented in Table 7 and Figure 3.

For the four endogenous variables, 82.5% of the variance of BI was explained by ATU, SE, and FC. ATU accounted for 22.7% of the variance by PU and PEU. PU was shown to have 13.3% variance by PEU and SN. For the PEU, its 25.4% variance rate was accounted for by SE, TC and FC.

## 5. Discussion

The empirical study explored the factors influencing Chinese English language teachers’ behavioral intentions to use online teaching during pandemic normalization. The relations among the factors were examined by developing a research model based on an extended TAM, including PU, PEU, and ATU as original variables from TAM, as well as SN, SE, TC, and FC as external variables.

### 5.1. Supported Relationships

#### 5.1.1. Variables within TAM

In line with the relationships explained in the models of technology acceptance [46,47], a significantly positive effect of ATU on BI was found in the study (H1). The finding was consistent with other empirical studies [5,75,76], which showed the significant role that ATU plays in E-leaning use or technology adoption. It is reasonable to infer that when English language teachers have positive feelings toward online teaching, their intentions to adopt online teaching will be reinforced. Therefore, improving English language teachers’ positive feelings toward online teaching might be a consideration for policymakers and institutional administrators who hope for teachers to engage more in online teaching.

ATU toward use was significantly influenced by PU and PEU (H3 and H4), suggesting that when the implementation of online teaching was perceived to be an enhancement to English language teachers’ performance and was relatively free of effort, English language teachers would develop a positive attitude toward its use. The findings highlighted that the benefits and advantages of online teaching should be demonstrated to English language teachers in the pandemic normalization [77]. Meanwhile, the findings also confirmed previous research, suggesting that system developers could collaborate with teachers to design online systems with a friendly interface, flexibility, and general purposes to facilitate English language teachers’ attitudes towards the adoption of online teaching [78].

PU was significantly influenced by PEU (H5) which was consistent with previous studies (e.g., Salloum, et al. [79]). The findings emphasized that if online teaching was considered free of effort or normative in a professional context by English language teachers, the perceived usefulness of online teaching would be increased, supporting their language teaching. Therefore, teachers need to have enough time and opportunities to familiarize themselves with online teaching until they have had enough positive interactions with online teaching and perceive that online teaching is useful in their language teaching [22].

#### 5.1.2. SN Variable

Regarding the external variables of TAM, the statistical results showed that SN exterted a significantly positive effect on PU (H7). The findings illustrated that, to some extent, the extent to which English language teachers perceived online teaching to be useful depended on the opinions of some influential people around them. The result was in line with previous studies (e.g., [80]). In the study investigating espoused cultural values’ influence on Chinese teachers’ intentions to use educational technology, SN was an important construct in predicting PU. This result is quite possible in the current study context. As mentioned above, English language teachers were not active to the incorporation of educational technology into their language teaching [81]. Therefore, they were concerned when online teaching was only one alternative to traditional teaching, especially in emergent crises. Teachers’ unfamiliarity with online teaching explained the statistical significance of the effect of SN on PU during the COVID-19 pandemic and in pandemic normalization. English language teachers’ online teaching practices, such as teaching strategies, teaching design, and organizing effective activities related to online teaching, were more likely to be affected by their peers, policies, professional training, and some experts from this field within a more regulated environment.

#### 5.1.3. SE Variable

SE was found to be significant in determining BI (H8) in an online teaching context according to the statistical results. The findings demonstrated that English language teachers who possessed a higher confidence in their ability to conduct online teaching would be more prone to adopting online teaching. The finding supported previous studies (e.g., Joo et al. [22]), while it was was not in line with all previous findings (e.g., Mei et al. [12]), which showed a non-significant influence of computer SE on preservice teachers’ acceptance of computer-assisted language learning technology. In addition, SE was significant in determining English language teachers’ PEU of online teaching (H9). This indicated that the more confidence English language teachers had in their perceived abilities regarding online teaching, the more easily English language teachers would accept online teaching. As the impact of SE on BI shown in previous studies varied based on the different research contexts, respondents, and cultures, the current findings supported some research results, such as those of Thongsri et al. [82], while contradicting other results, such as those of Al Kurdi et al. [83]). After extensively exploring online teaching with support from different aspects, English language teachers’ confidence in online teaching has been enhanced. However, the findings still provide a reminder that further studies should be conducted to explore the factors that could improve English language teachers’ SE.

#### 5.1.4. FC Variable

In line with the previous studies of Lin et al. [57,62], FC was shown to have a significantly positive effect on BI (H11) and PEU (H12). The findings suggested that English language teachers’ perception of adequate support affected the extent to which online teaching was perceived to be free of effort and teachers’ willingness to conduct online teaching. Despite the lack of the specification of facilitating conditions in the study, a multiple-facet FC can be offered to English language teachers in pandemic normalization.

Firstly, timely technological support in an online setting is critical to avoid panic or chaos caused by emergencies or bad interactions between teachers and online teaching media. Therefore, the efficient and effective use of technology necessitates assistance at all levels of the organization [84]). Knowledge and skills were also critical to the successful implementation of online teaching. The finding of Zheng et al. [85] showed that teachers would be more confident in using technology if they had sufficient knowledge and skills. These knowledge and skills can partially be acquired through professional development training, technological support, and the exchange of experiences among peers. Here, the focus on professional development programs offered by higher education institutions is necessary for administrators to help teachers obtain the necessary knowledge and skills for online teaching. The previous study has shown that administrators’ commitment to addressing teachers’ fear of online teaching can motivate faculty to engage in online courses [86]. In addition, the study by Bin [87] showed that a lack of incentive and reward systems could restrict the benefits attained by the deployment of the learning system. Therefore, some allowance is essential to increasing English language teachers’ involvement in time-consuming online teaching.

### 5.2. Unsupported Relationships

Three hypotheses out of twelve were not supported according to the research results, including PU→BI, SN→BI, and TC→PEU.

#### 5.2.1. PU→BI

The result that PU did not significantly influence BI (H2) was surprising and contrary to their relationship with the original TAM [13] from the theory perspective and other empirical studies (e.g., Rafique et al. [88], while it was consistent with some studies (e.g., Mensah et al. [80]). The lack of a significant impact of PU on BI indicated that English language teachers’ behavioral intentions to adopt online teaching was not affected by their perceived usefulness. However, it is reasonable to obtain this result in the pandemic setting. For English language teachers, the first consideration when conducting online teaching was not its usefulness in improving their performance in English teaching but their having no option to continue normal educational activities during the COVID-19 pandemic. As online English teaching was perceived to be relatively complex, it was also possible that PU was not a powerful enough driver to affect English language teachers’ intention to adopt online teaching in pandemic normalization if other factors, such as perceived ease of use or attitudes toward online teaching, were not present.

#### 5.2.2. SN→BI

The findings showed that SN did not significantly affect BI (H6). Although PU was affected by others, English language teachers’ willingness to adopt online teaching did not depend on the perceptions of some influential people. Instead, they made their own evaluations and established their own preference regarding whether they should engage in online teaching. This finding was not consistent with other studies, such as that of Ursavaş et al. [76,89]). In these studies, SN was a significant determinant of behavioral intentions. For example, in the study of Ursavaş et al., SN exerted a large influence on preservice teachers’ intentions to use technologies. It is possible to explain the result from the perspective of demographics. Most participants (71.1%) in the study had more than 10 years of teaching experience; therefore, their decision about whether they adopted online teaching or not would be less likely to be affected by the opinion of others. Another possible reason is that, although conducting online teaching was mandated by institutional policies during the pandemic, the impact of external factors would not be effective when deciding whether to continue with online teaching.

#### 5.2.3. TC→PEU

TC had a negative but not significant effect on PEU (H10) in the study, which revealed that the perceived ease of use of online teaching was not affected by whether the technology was complex. Although this is contrary to our expectations and previous studies (e.g., Teo et al. [18]), it is possible that, in the current study, English language teachers who had gone through the emergent pandemic’s requirements for remote teaching increased their ability and confidence in how to conduct an online course via online trainings from technical staff, professional workshops from field experts, shared experience from their peers, and support from institutions. Therefore, they may not consider online teaching to be as complex in pandemic normalization after this personal experience with online teaching. Additionally, English language teachers who are exposed to digital technologies may think that online teaching is a common educational mode and not have difficulties adopting it. Therefore, the lack of a significant effect of TC on PEU was reasonable in the study.

## 6. Limitations and Implications

Although the empirical study was carefully designed, some limitations existed. First, considering the huge imbalance in the development of economics and education among regions in China, only English language teachers from one province were selected in this tentative study. In other words, the findings of the study should be generalized with caution. Therefore, future studies could be conducted with a larger sample size from different areas to conduct some comparative analysis and explore a more comprehensive picture of English language teachers’ intentions regarding online teaching. Second, some uncontrollable factors, such as respondents’ subjective emotions and social expectations, will affect the quality of data due to the use of a self-report questionnaire. Therefore, some other types of data can be collected, such as qualitative data, to explore comprehensive phenomena for similar topics.

Despite some limitations, the study has several implications from the perspective of theories and practice. First of all, the study verified TAM’s wide application in the domain (online teaching) and context-specific constructs (in China). In addition, the unsupported impact of PU on BI enriched the TAM study, as opposed to the commonly accepted relations between PU and BI in the TAM. This means that the relationships between variables with a model do not remain static. Furthermore, the empirically identified relationships between these factors of expanding TAM suggest that technology acceptance does not only rely on the technology itself, but other factors, such as cognitive, social, and psychological factors, also play critical roles in an individual’s intention to perform a behavior.

The study has direct implications for stakeholders such as teachers, policymakers, leaders, and technology developers from the point of practice. For example, attitude and self-efficacy have a positive influence on English language teachers’ acceptance of online teaching; therefore, measures should be taken to ensure teachers have positive feelings about online teaching. As the PEU has a significant effect on ATU, when designing the system, it will be better for developers to improve the convenience of the operation of educational technologies and develop much simpler, more efficient, and universally applicable platforms, along with offering technological support. In addition, FC is the most important factor affecting BI in the study; hence, support and encouragement from leaders will improve teachers’ willingness to participate. Additionally, regular technological, pedagogical, and subject content training being offered by administrations is necessary to facilitate teachers’ successful online teaching.

## 7. Conclusions

The empirical study offered extensive insights into English teachers’ intentions to teach online during the pandemic normalization of COVID-19. The large variance in BI was explained by ATU, SE, and FC. FC has a stronger impact on English language teachers’ intentions to adopt online teaching than the other two factors in the current pandemic normalization setting. Therefore, determining the potentially specific facilitating conditions and providing them to teachers are important in times of crisis to improve English teachers’ willing to adopt online teaching in their practice. More importantly, potential influencing factors from different perspectives should be determined in future research. For example, mental factors such as stress and anxiety, which have been investigated in college students’ online learning during COVID-19 [90], can be incorporated into the model to examine their influence on teachers’ behavioral intentions regarding online teaching. It is still unclear whether and to what extent remote forms of instruction should replace traditional instruction, and whether the clear disadvantages of this line of work outweigh the benefits. Therefore, conducting similar studies with respect to those who are instructing in different academic fields would be quite fascinating.

## Figures and Tables

**Figure 1 behavsci-13-00624-f001:**
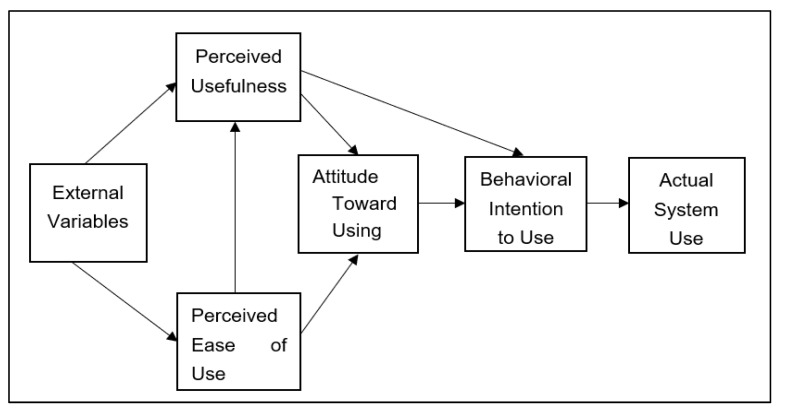
Theory of acceptance model [13].

**Figure 2 behavsci-13-00624-f002:**
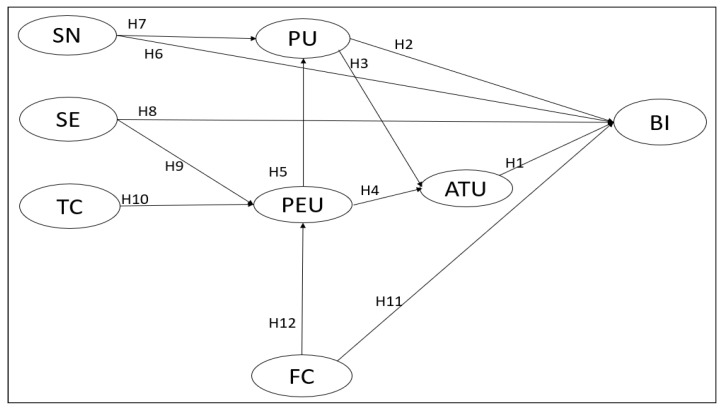
Proposed research model. Notes. BI: behavioral intention to adopt online teaching; ATU: attitude toward online teaching; PU: perceived usefulness; PEU: perceived ease of use; SN: subjective norm; SE: self-efficacy; TC: technological complexity; FC: facilitating conditions.

**Figure 3 behavsci-13-00624-f003:**
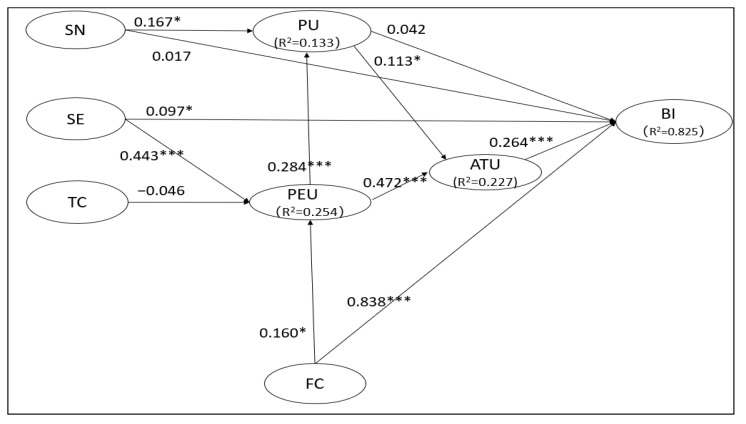
The results of structural model testing. Notes: * *p* < 0.05; *** *p* < 0.001; BI: behavioral intentions to adopt online teaching; ATU: attitude toward online teaching; PU: perceived usefulness; PEU: perceived ease of use; SN: subjective norm; SE: self-efficacy; TC: technological complexity; FC: facilitating conditions.

**Table 1 behavsci-13-00624-t001:** The profile of the participants (N = 287).

Variable	Classification	Frequency	Percent
Gender	Male	59	20.6
	Female	228	79.4
Age distribution	25–35	68	23.7
	36–45	136	47.4
	46–55	58	20.2
	Over 56	25	8.7
Teaching years	Under 5 years	42	14.6
	6–10 years	41	14.3
	11–15 years	65	22.6
	16–20 years	52	18.1
	Over 20 years	87	30.3
Academic title	Teaching Assistant	32	11.1
	Lecturer	130	45.3
	Associate Professor	93	32.4
	Professor	32	11.1
Education level	Bachelor	24	8.4
	Master	225	78.4
	PhD	38	13.2

**Table 2 behavsci-13-00624-t002:** The instruments’ information.

Construct	Sources	Number of Items	Likert Scale	Cronbach’s Alpha	
				Adopted	Pilot
BI	[13]	3	5-point	0.970	0.932
ATU	[28]	3	5-point	0.972	0.829
PU	[13]	3	5-point	0.920	0.920
PEU	[13]	3	5-point	0.921	0.862
SN	[55]	3	5-point	0.860	0.890
FC	[55]	3	5-point	0.865	0.884
SE	[60]	3	5-point	0.970	0.846
TC	[55]	3	5-point	0.936	0.860

Notes. BI = behavioral intentions to adopt online teaching; ATU = attitude toward online teaching; PU = perceived usefulness; PEU = perceived ease of use; SN = subjective norm; SE = self-efficacy; TC = technological complexity; FC = facilitating conditions.

**Table 3 behavsci-13-00624-t003:** Summary of fit indices of measurement mode.

Fit Indices	Recommended Criteria	Results of the Research Model
CMIN (λ^2^)	Smaller is better	380.199
df	Bigger is better	224
λ^2^/df	<3	1.697
CFI	>0.9	0.963
TLI	>0.9	0.955
RMSEA	<0.08	0.049
SRMR	<0.08	0.043

Notes. CMIN = Chi-square value; df = degree of freedom; λ^2^/df = the ratio of CMIN and its degree of freedom; CFI = comparative fit index; TLI = Tucker–Lewis’s index; RMSEA = root mean square error of approximation; SRMR = standardized root mean square residual.

**Table 4 behavsci-13-00624-t004:** Summary of the measurement model results.

Construct	Indicator	Sig. Test of Parameters				Std.	Item Reliability	Composite Reliability	Convergence
		Unstd.	S.E.	*t*-Value	*p*		SMC	CR	AVE
BI	BI1	1.000				0.836	0.699	0.872	0.695
	BI2	1.144	0.065	17.582	***	0.866	0.750		
	BI3	0.940	0.060	15.637	***	0.798	0.637		
ATU	ATU1	1.000				0.779	0.607	0.820	0.604
	ATU2	0.783	0.069	11.277	***	0.710	0.504		
	ATU3	1.002	0.082	12.249	***	0.837	0.701		
PU	PU1	1.000				0.702	0.493	0.853	0.661
	PU2	1.019	0.082	12.480	***	0.827	0.684		
	PU3	1.220	0.097	12.610	***	0.898	0.806		
PEU	PEU1	1.000				0.811	0.658	0.881	0.713
	PEU2	1.108	0.065	17.005	***	0.920	0.846		
	PEU3	0.954	0.064	14.944	***	0.796	0.634		
SN	SN1	1.000				0.862	0.743	0.885	0.719
	SN2	0.948	0.058	16.418	***	0.808	0.653		
	SN3	0.962	0.052	18.332	***	0.873	0.762		
SE	SE1	1.000				0.845	0.714	0.840	0.637
	SE2	0.901	0.063	14.228	***	0.832	0.692		
	SE3	0.844	0.068	12.379	***	0.711	0.506		
TC	TC1	1.000				0.751	0.564	0.818	0.602
	TC2	1.245	0.090	13.869	***	0.859	0.738		
	TC3	1.004	0.087	11.593	***	0.709	0.503		
FC	FC1	1.000				0.840	0.706	0.852	0.658
	FC2	0.978	0.060	16.339	***	0.812	0.659		
	FC3	0.885	0.057	15.416	***	0.781	0.610		

Notes. BI = behavioral intentions to adopt online teaching; ATU = attitude toward online teaching; PU = perceived usefulness; PEU = perceived ease of use; SN = subjective norm; SE = self-efficacy; TC = technological complexity; FC = facilitating conditions; Unstd. = unstandardized estimates; S.E. = standard error; Std. = standardized estimates; SMC = square multiple correlations; CR = composite reliability; AVE = average variance extracted; *** *p* < 0.001.

**Table 5 behavsci-13-00624-t005:** Assessment of discriminant validity.

	SN	FC	TC	SE	PEU	PU	ATU	BI
SN	**0.848**							
FC	0.747	**0.811**						
TC	0.699	0.754	**0.776**					
SE	0.366	0.409	0.380	**0.798**				
PEU	0.258	0.302	0.259	0.491	**0.844**			
PU	0.241	0.220	0.191	0.201	0.327	**0.813**		
ATU	0.125	0.145	0.125	0.234	0.476	0.224	**0.777**	
BI	0.736	0.702	0.774	0.457	0.323	0.252	0.305	**0.834**

Notes. BI = behavioral intentions to adopt online teaching; ATU = attitude toward online teaching; PU = perceived usefulness; PEU = perceived ease of use; SN = subjective norm; SE: self-efficacy; TC = technological complexity; FC = facilitating conditions; diagonal elements in bold are the square root of the AVE.

**Table 6 behavsci-13-00624-t006:** Summary of fit indices of structural model.

Fit Indices	Recommended Criteria	Results of the Research Model
CMIN (λ^2^)	Smaller is better	505.953
df	Bigger is better	234
λ^2^/df	<3	2.162
CFI	>0.9	0.936
TLI	>0.9	0.924
RMSEA	<0.08	0.064
SRMR	<0.08	0.092

Notes. CMIN (λ^2^) = Chi-square value; df = degree of freedom; λ^2^/df = the ratio of CMIN and its degree of freedom; CFI = comparative fit index; TLI = Tucker–Lewis’s index; RMSEA = root mean square error of approximation; SRMR = standardized root means square residual.

**Table 7 behavsci-13-00624-t007:** Summary of structure model.

Hypotheses	Path	Path Coefficient	*t*-Value	Results
H1	Attitude → Behavioral intentions	0.264 ***	4.013	Supported
H2	Perceived usefulness → Behavioral intentions	0.042	1.008	Not supported
H3	Perceived usefulness → Attitude	0.113 *	2.083	Supported
H4	Perceived ease of use → Attitude	0.472 ***	6.421	Supported
H5	Perceived ease of use → Perceived usefulness	0.284 ***	4.125	Supported
H6	Subjective norm → Behavioral intentions	0.017	0.195	Not supported
H7	Subjective norm → Perceived usefulness	0.167 *	2.517	Supported
H8	Self-efficacy → Behavioral intentions	0.097 *	1.971	Supported
H9	Self-efficacy → Perceived ease of use	0.443 ***	6.066	Supported
H10	Technological complexity → Perceived ease of use	−0.046	−0.298	Not supported
H11	Facilitating conditions → Behavioral intentions	0.838 ***	8.560	Supported
H12	Facilitating conditions → Perceived ease of use	0.160 *	2.247	Supported

Notes: * *p* < 0.05; *** *p* < 0.001.

## Data Availability

Data supporting the reported results are available on request.

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
