# Peer review of "Factors Affecting English Language Teachers’ Behavioral Intentions to Teach Online under the Pandemic Normalization of COVID-19 in China"

_behavsci, 2023, doi:10.3390/bs13080624_

Round 1

Reviewer 1 Report

As it claims, the study examined the factors that affected English language teachers’ behavioral intentions to teach online under the normalization of the COVID-19 pandemic in China. Here is my feedback:

1. Title: I don’t think the word ‘modelling’ would properly describe this study. I had a different expectation on my mind the first time I read the title—not SEM for sure.

2. Abstract: The authors should include a concise statement about the research gap/ the rationale behind the study, etc. before explaining the objective of the study (the first sentence). Explain where the questionnaire came from and if any validating processes were involved. Add the details of % male and % female for the respondent. The last sentence should be elaborated on briefly. I know this kind of promise is common for some, but you should just outline what they are briefly in the abstract.

3. Introduction: Look at the title, then read the first and second paragraphs. Do they match well? I think they don’t. I think your third paragraph should be the first paragraph. You need to reorganize the paragraphs in your introduction. Make it directly talk about the focus of the research. You can begin with the online teaching during the outbreak and the teacher’s intention to teach online. Present what has been explored and what has not been explored. Then, talk about TAM and how it can be used to measure teachers' attention to teaching online. At this point, I am not convinced that TAM can be used for that purpose. There is no need to talk about how ICT impacts our lives, COVID infections, etc. Be direct and focused on the topic of your research. For this, you will need to make extensive revisions to your introduction.

4. Literature review: I learned a lot about TAM and the included variables, but I can’t see how they would work with measuring a teacher’s intention to teach online. The simple thing about using TAM, as the acronym stands for Technology Acceptance Model, is that there should be a specific technology at the center of whether it should be accepted, but this study does not specify any. I am confused. The authors do not have clear explanations connecting TAM and the included variables to the teacher’s intention to teach online. In fact, during the outbreak, did they have a choice other than teaching online? Unless they or their schools do not have the appropriate platforms, they had to teach classes online, didn’t they? Unless they wanted to lose their jobs or get infected by COVID. This is the most difficult part to understand. The manuscript does not provide a review of studies regarding teachers’ intentions to teach online during the outbreak, so readers can’t see what the problem is.

5. Methodology:

The methodology section is well-explained and comprehensive.

6. Results:

The discussion section is well-written; however, since there is limited literature review on teachers' intentions to teach online during the pandemic, the study makes certain assumptions based on the observed correlations among the included variables. To strengthen the discussion, it is recommended to include more comprehensive reviews of teachers' intentions to teach online during the outbreak, which would provide a better context for interpreting the results.

Reviewer 2 Report

I find the research interesting for the scientific community. The research approach is well structured. The hypotheses are consistent with the general objective of the investigation. Through the research findings, interesting conclusions can be drawn according to the influence of self-efficacy, attitude and facilitating conditions. The authors could briefly relate some conclusions to future lines of research according to the level of stress: an article that could be of use would be: Spain and Costa Rica during Periods of Confinement and Virtual Learning.

The "figure 2 Proposed research model" should be put more clearly. Figure 2 does not look good.

The article can be published when the authors make all the indicated suggestions.

Reviewer 3 Report

The early 2020 events, which were brought on by the sudden and unexpected spread of the SARS CoV-2 virus that had global pandemic-like magnitude, have created challenging circumstances in all spheres of humanity's endeavors. The widespread introduction of new distant education formats and methods of knowledge transfer and enforcement has raised a number of challenges as well as new potential of improving education conditions. The newly established scenario has forced the need of quickly adapting to completely new and not yet recognized conditions. I think the paper to be intriguing and up to date from this perspective. It should also be emphasized that the researched problem has been addressed in an exhaustive manner.

The findings of the research are included in the ongoing scientific debate about changes to the educational system in the world's scientific literature, which has become more intense in recent years, particularly in the aftermath of the COVID-19 pandemic. Conclusions derived from the analysis of the respondents' opinions can offer some guidance for a new method of teaching for the digital age, but it is still unclear whether and to what extent remote forms of instruction should replace traditional instruction and whether the clear disadvantages of this line of work do not outweigh the benefits. I believe that conducting similar studies with respect to those who are instructing different academic fields would be quite fascinating.

Although I am aware that the article is quite extensive, I would simply like to draw attention to the fact that there was a slight absence of references to the findings of similar studies from other parts of the world, which I believe would have contributed to the article's content.

Round 2

Reviewer 1 Report

Dear authors, 

Thank you for your revision. I only have concern at the moment. You wrote these in the abstract: 

"Teachers play a significant role in any successful teaching practices. However, previous studies showed that English teachers in China were negative or reluctant to conduct online teaching 16 during COVID-19 pandemic."

You will need to revise these, especially the first sentence. It does not make any sense. Of course, teachers play a significant role in successful teaching practises because they are the ones teaching; who else? And the first and second sentences, as shown here, are not well connected. You will need to revise these two sentences and change them into something that will emphasise the urgency of your research.

Congratulations! I hope to see your article published soon!
